# Nicotinic Acid Catabolism Modulates Bacterial Mycophagy in *Burkholderia gladioli* Strain NGJ1

Joyati Das,[a] Rahul Kumar,[a] Sunil Kumar Yadav,[a] (ORCID) Gopaljee Jha[a]

[a]Plant Microbe Interactions Laboratory, National Institute of Plant Genome Research, New Delhi, India

**ABSTRACT** *Burkholderia gladioli* strain NGJ1 exhibits mycophagous activity on a broad range of fungi, including *Rhizoctonia solani*, a devastating plant pathogen. Here, we demonstrate that the nicotinic acid (NA) catabolic pathway in NGJ1 is required for mycophagy. NGJ1 is auxotrophic to NA and it potentially senses *R. solani* as a NA source. Mutation in the *nicC* and *nicX* genes involved in NA catabolism renders defects in mycophagy and the mutant bacteria are unable to utilize *R. solani* extract as the sole nutrient source. As supplementation of NA, but not FA (fumaric acid, the end product of NA catabolism) restores the mycophagous ability of Δ*nicC*/Δ*nicX* mutants, we anticipate that NA is not required as a carbon source for the bacterium during mycophagy. Notably, *nicR*, a MarR-type of transcriptional regulator that functions as a negative regulator of the NA catabolic pathway is upregulated in Δ*nicC*/Δ*nicX* mutant and upon NA supplementation the *nicR* expression is reduced to the basal level in both the mutants. The Δ*nicR* mutant produces excessive biofilm and is completely defective in swimming motility. On the other hand, Δ*nicC*/Δ*nicX* mutants are compromised in swimming motility as well as biofilm formation, potentially due to the upregulation of *nicR*. Our data suggest that a defect in NA catabolism alters the NA pool in the bacterium and upregulates *nicR* which in turn suppresses bacterial motility as well as biofilm formation, leading to mycophagy defects.

**IMPORTANCE** Mycophagy is an important trait through which certain bacteria forage over fungal mycelia and utilize fungal biomass as a nutrient source to thrive in hostile environments. The present study emphasizes that nicotinic acid (NA) is important for bacterial motility and biofilm formation during mycophagy by *Burkholderia gladioli* strain NGJ1. Defects in NA catabolism potentially alter the cellular NA pool, upregulate the expression of *nicR*, a negative regulator of biofilm, and therefore suppress bacterial motility as well as biofilm formation, leading to mycophagy defects.

**KEYWORDS** bacterial-fungal interaction, bacteriology, *Burkholderia gladioli*, environmental microbiology, genetics and molecular biology, mycophagy, nicotinic acid, biofilms, swimming motility

Bacteria and fungi often form interlinked communities in the environment and exhibit antagonistic or synergistic interactions. Some of them possess potent antifungal activity, while a few others exhibit a unique property to forage over fungal mycelia, a phenomenon known as bacterial mycophagy (1). One of the classical examples of mycophagous bacteria is *Collimonas* sp. which feeds on some of the soil-habiting fungi, particularly under nutrient-limiting conditions (2). A rice endophytic bacterium, *Burkholderia gladioli* strain NGJ1 demonstrates mycophagous activity on a broad-range of fungi, including *Rhizoctonia solani*, an agriculturally important polyphagous fungal pathogen (3–5). It has been anticipated that during bacterial mycophagy, lysis of the fungal cells leads to the release of fungal metabolites into the extracellular environment and makes them accessible to the bacterial partner (6). For example, the bacterium *B. terrae* feeds on *Lyophyllum* sp. strain Karsten and utilizes the released fungal

Address correspondence to Gopaljee Jha, jmsgopal@nipgr.ac.in, or jmsgopal@gmail.com.

The authors declare no conflict of interest.

metabolite (glycerol) as an energy source (7). The underlying mechanism by which the mycophagous bacteria perceive fungi remains elusive. Generally, quorum sensing (QS) molecules are likely used as cues during bacterial-fungal interactions (8, 9). It has been anticipated that bacteria may sense some of the soluble compounds released by the fungal partner. For example, *Collimonas* bacterium exhibits chemotactic behavior toward oxalic acid, secreted by soil fungi (10). However, it is not clear how *B. gladioli* NGJ1 senses fungi.

Nicotinic acid (NA, a pyridine derivative) is vitamin B3, and it serves as a part of some of the important cofactors (such as NAD and NADP) in bacteria (11, 12). Some of the bacteria can utilize NA as a carbon or nitrogen source (11, 13). The bacteria that cannot synthesize NA, need to obtain it from external sources (14). For example, *Bordetella* sp., an animal pathogenic bacterium is auxotrophic for NA (15). NA catabolic pathway has been elaborately characterized in *Pseudomonas putida* KT2440 (16). The *nic* gene clusters, organized in different operons *viz.*, *nicAB*, *nicCDEFTP*, and *nicXR*, are involved in the catabolism of NA to fumaric acid (FA), a TCA cycle intermediate in *P. putida*. During catabolism, NA is initially converted into 6-hydroxynicotinic acid (6HNA) by the two-component hydroxylase, NicA and NicB. Thereafter, 6HNA is converted to 2,5-dihydroxypyridine (2,5DHP) by the enzyme 6-hydroxynicotinate 3-mono-oxygenase encoded by *nicC* gene. 2,5DHP is further converted into FA by enzymes encoded by other *nic* genes, such as *nicX*, *nicD*, *nicF*, and *nicE*. As depletion of NA pool adversely affects the NAD homeostasis in the cells, its degradation pathway is tightly regulated in bacteria (14). In *P. putida*, a MarR-type transcriptional repressor, NicR represses the *nicC* and *nicX* operons, whereas NicS represses the *nicAB* genes (17). In *B. bronchiseptica*, BpsR (NicR homolog) controls bacterial growth by repressing the *nic* genes (15). Mutational analysis revealed that dysregulation of the *nic* genes adversely affects the growth as well as the virulence of the bacteria (15).

In the present study, we report that disruption in NA catabolic pathway genes (*nicC*/*nicX*) compromises the mycophagous ability of *B. gladioli* NGJ1 against *R. solani*. The supplementation of NA promotes mycophagy in the Δ*nicC*/Δ*nicX* mutant bacteria. Our data suggest that defects in NA catabolism potentially alter the NA pool and up-regulate *nicR*, which suppresses bacterial motility as well as biofilm formation and renders the Δ*nicC*/Δ*nicX* mutant bacteria defective in mycophagy.

## RESULTS

**Transposon mutagenesis screen identified *nicC* gene to be important for mycophagous ability of *B. gladioli* strain NGJ1.** We carried out *Tn5gusA11*-based transposon mutagenesis in *B. gladioli* NGJ1 to identify gene(s) required for bacterial mycophagy. Out of 84 mutant strains that were transposon-tagged (GUS positive), 12 were defective in mycophagy against *R. solani*, at 7 DPI (Fig. S1A). We selected one of them (NGJ1_6) wherein the *Tn5gusA11* insertion was in the *nicC* gene (ACI79_RS00345; Burkholderia genome database Locus ID; https://www.burkholderia.com/) that encodes 6-hydroxynicotinate 3-monooxygenase, an important enzyme in nicotinic acid (NA) catabolism (16) for further characterization. Notably, the NGJ1_6 was compromised in foraging over *R. solani* mycelia, even at 7 DPI (Fig. S1B). On the other hand, wild-type NGJ1 foraged over fungal mycelia, at both 3 and 7 DPI (Fig. S1).

***nicC* gene is a part of NA catabolic gene cluster in *B. gladioli* strain NGJ1.** Previous studies had identified various genes associated with NA catabolic (nic) pathway in *Pseudomonas putida* (16) and *Bordetella bronchiseptica* (15). As the KEGG pathway of NGJ1 is not available in the database, we used the genome sequence of *B. gladioli* strain BSR3 (18) to identify NA catabolic pathway genes (Fig. S2). Similar to *B. bronchiseptica* and *P. putida*, most of the genes associated with NA catabolism are encoded in an operonic fashion in *B. gladioli* strain BSR3 (Fig. 1A). BLAST analysis revealed that *B. gladioli* strain NGJ1 also harbors a complete *nic* gene cluster (Table S1, Fig. 1A).

**NA catabolic genes (*nicC* and *nicX*) are required for bacterial mycophagy by *B. gladioli* strain NGJ1.** In order to validate whether mycophagy defect in NGJ1_6 is due to transposon-mediated disruption of the *nicC* gene, we created Δ*nicC*, an

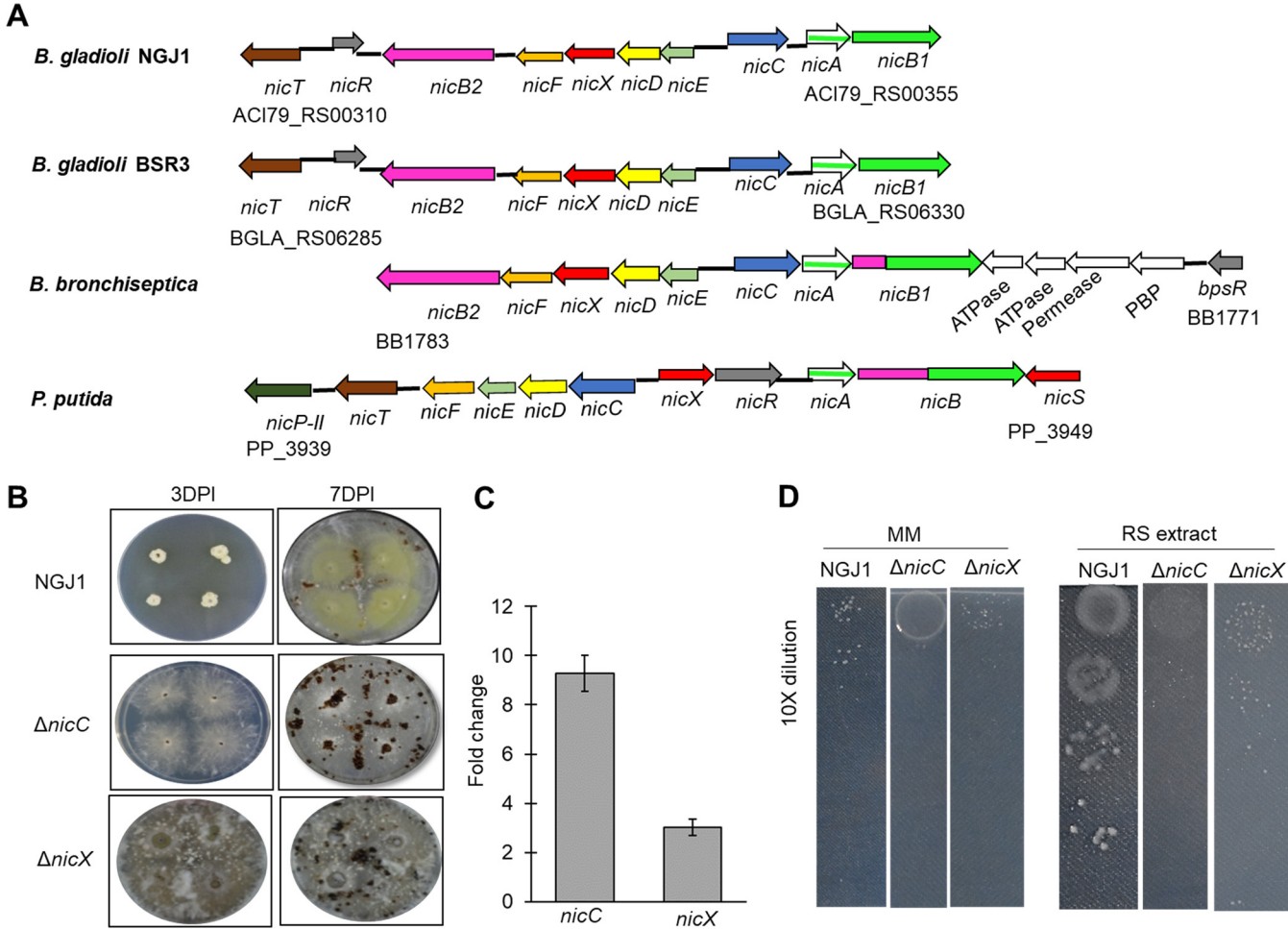

**FIG 1** Mutation in *nicC* and *nicX* genes involved in NA catabolic pathway imparts mycophagy defects in NGJ1 against *R. solani*. (A) Genomic organization of *nic* genes in *B. gladioli* (NGJ1 and BSR3), *B. bronchiseptica* RB50 and *P. putida* KT2440. The locus ID of the beginning and end of the gene cluster is provided. (B) The interaction of *B. gladioli* NGJ1 strains ($10^3$ cells/mL density) with *R. solani* on PDA plates at 3 and 7 DPI. (C) qRT-PCR-based expression of *nicC* and *nicX* genes of NGJ1 during 48h of mycophagous interaction with *R. solani*. The fold change was estimated by $2^{-\Delta\Delta Ct}$ method using 16S rRNA of NGJ1 as the endogenous control. (D) Growth of serially diluted bacterial cultures on M9 minimal media (MM), with and without supplementation of *R. solani* mycelial extract (RS extract). The experiments were independently repeated three times with a minimum of three technical replicates. Graphs show mean values ± standard error of three biological replicates.

independent insertional mutant in NGJ1. Similar to the transposon-tagged mutant, the Δ*nicC* was deficient in bacterial mycophagy on *R. solani* (Fig. 1B). Considering that *nicC* is a part of the *nic* operon (Fig. 1A), we envisaged that insertion in the gene may lead to disruption of downstream genes associated with NA catabolism. Hence, to explore the importance of NA catabolic pathway during mycophagy, we investigated the involvement of a downstream gene i.e. *nicX* (2,5-dihydroxypyridine dioxygenase; ACI79_RS00335). The qRT-PCR analysis reflected that both *nicC* and *nicX* were upregulated during mycophagous interaction with *R. solani* (Fig. 1C). We created an insertion mutant in the *nicX* gene and observed that mutant strain (Δ*nicX*) was defective in foraging over *R. solani* (Fig. 1 B). The bacterial abundance (log CFU/mL) on *R. solani* mycelia was negligible during the confrontation of Δ*nicC* and Δ*nicX*, while abundant bacterial growth on the mycelia was observed in case of wild-type NGJ1 (Fig. S3 A). Further, we used a colorimetric MTT (3-[4,5-dimethylthiazol-2-yl]-2,5-diphenyltetrazolium bromide) assay to detect the viability of bacterial-treated *R. solani* mycelia. It is known that viable cells convert MTT into a colored compound, while nonviable cells remain colorless (19). The treatment with Δ*nicC* and Δ*nicX* mutant bacterial strains failed to induce cell death in *R. solani* mycelia, as evident by the formation of color (Fig. S3 B). On the other hand, the wild-type NGJ1-treated *R. solani*

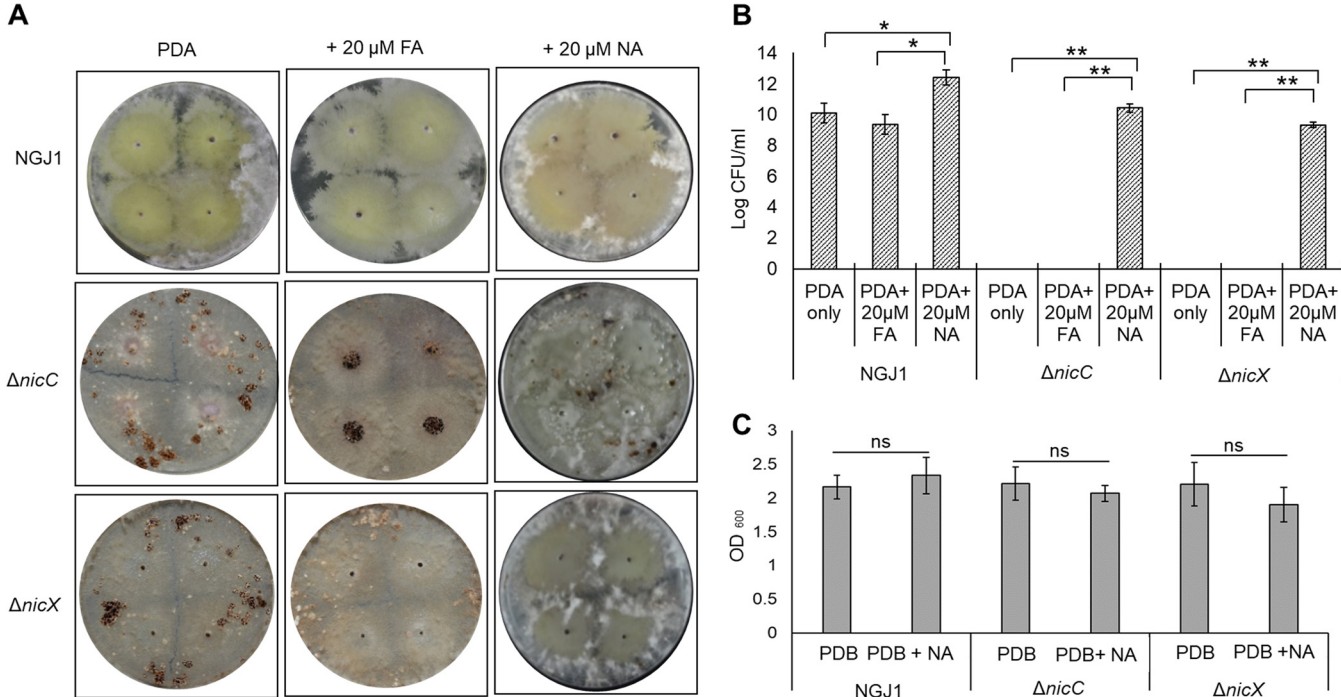

**FIG 2** NA complements mycophagy defects associated with ΔnicC and ΔnicX mutants. (A) Representative images of the interaction of NGJ1 strains ($10^3$ cells/mL density) with *R. solani* (RS) on PDA plates, with and without supplementation of fumaric acid (FA) or nicotinic acid (NA), at 7 DPI. (B) Bacterial abundance on fungal mycelia after 7 days of confrontation. (C) The optical density ($OD_{600}$) of bacterial cultures grown in PDB media (24 h), in the presence or absence of NA. The experiments were independently repeated three times with a minimum of three technical replicates. Graphs show mean values ± standard error of three biological replicates. Asterisks * and ** indicate a significant difference at $P < 0.05$ and $P < 0.01$, respectively (estimated using one-way ANOVA). NS, nonsignificant.

mycelia were colorless, emphasizing that they were nonviable. Further, the spectrophotometric measurement reflected that ΔnicC and ΔnicX mutant treated mycelia had significantly more formazon, compared to those treated with wild-type bacterium (Fig. S3 C). Overall, this suggested that disruption in either *nicC* or *nicX* genes of NA catabolic pathway compromises NGJ1 to forage over *R. solani*.

**NA is required for the growth of *B. gladioli* strain NGJ1 in minimal media.** We observed that the wild-type as well as ΔnicC and ΔnicX mutant NGJ1 strains were unable to grow in M9 minimal media that is devoid of any carbon source, whereas the bacterial growth was comparable in rich media (PDB) (Fig. S3 D). The supplementation of NA restored the growth of wild-type NGJ (to a certain extent), whereas other N-heterocyclic aromatic compounds i.e. picolinic acid (PA) and isonicotinic acid (INA) failed to promote bacterial growth in minimal media (Fig. S3 D). In contrast, the addition of NA/N-heterocyclic aromatic compounds was unable to support the growth of ΔnicC and ΔnicX mutant strains (Fig. S3 D). Moreover, we observed that supplementation of cell extract of *R. solani*, promoted the growth of wild-type but not ΔnicC and ΔnicX mutant strains (Fig. 1D). Taken together, our data suggest that *B. gladioli* strain NGJ1 may sense *R. solani* as a source of NA, while ΔnicC/ΔnicX mutants are defective in this process.

**Supplementation of NA restores mycophagy in ΔnicC and ΔnicX mutants of *B. gladioli* strain NGJ1.** Considering that the end product of NA catabolism is fumaric acid (FA), a TCA cycle intermediate, we tested whether supplementation with FA (20 μM) restores mycophagy in the ΔnicC/ΔnicX mutants. The analysis revealed that ΔnicC and ΔnicX are still compromised in mycophagy against *R. solani* on FA supplemented plates (Fig. 2A). On the other hand, supplementation of NA (20 μM), restored mycophagy in both ΔnicC as well as ΔnicX mutant strains (Fig. 2A), and the bacterial abundance on fungal mycelia was enhanced (Fig. 2B). It is to be noted that supplementation of NA has no apparent effect on the growth of the wild-type, ΔnicC, and ΔnicX

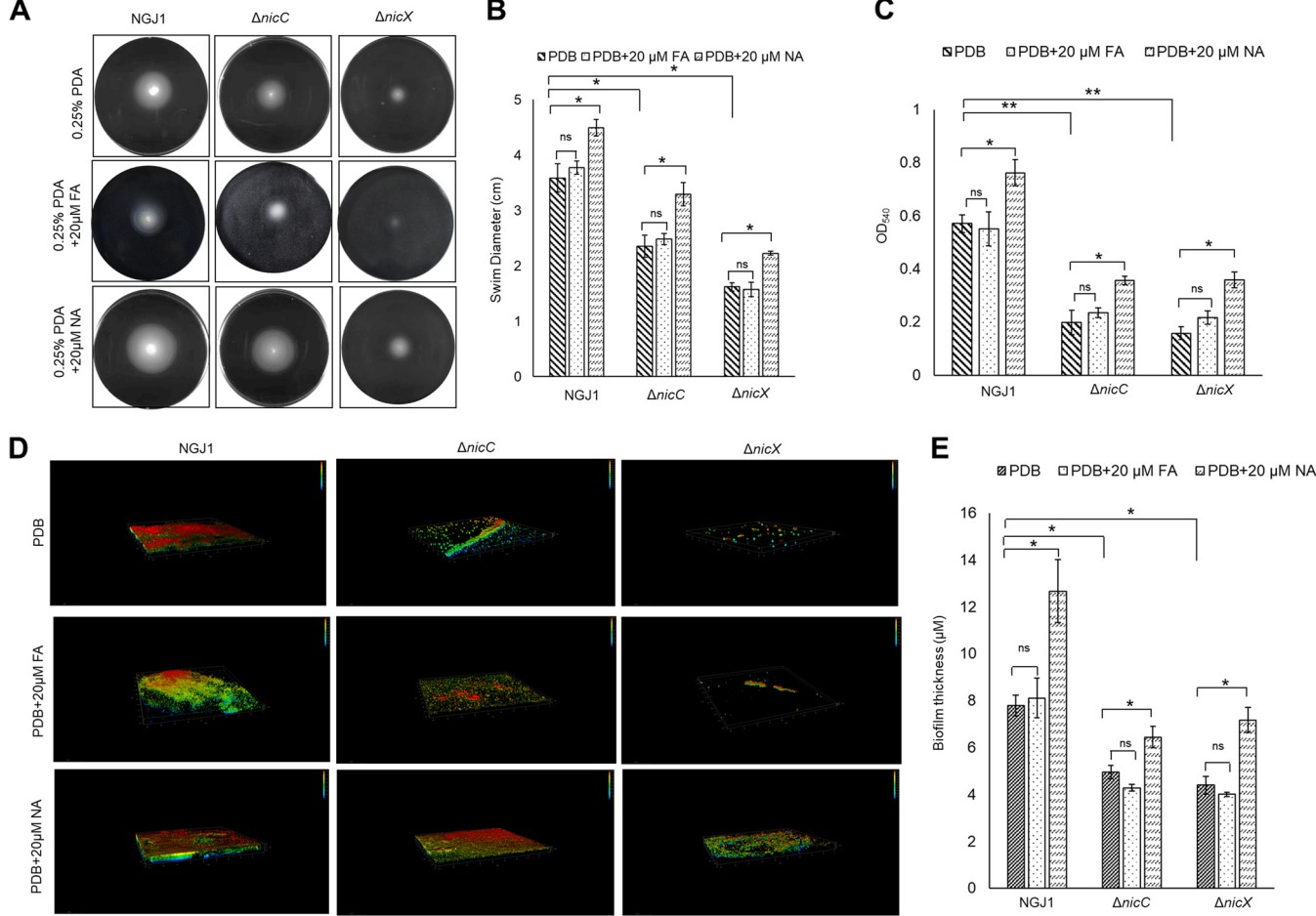

**FIG 3** Δ*nicC* and Δ*nicX* mutant strains are defective in swimming motility and biofilm formation. (A) Representative images showing swimming motility of NGJ1 strains after 16 h of growth on semisolid swim (0.25% PDA) plates. (B) The diameter of the motility zone on the swim plates. (C) Spectrophotometric quantification of crystal violet ($OD_{540}$) extracted from the surface-attached bacterial cells, upon 72 h of growth in polystyrene plates, reflecting the number of cells forming static biofilm. (D) Representative confocal laser-scanning microscopic (CLSM) images showing biofilm formed by different NGJ1 strains on glass slides at the air-media interface upon staining with BacLight LIVE/DEAD stain during 72 h growth. Each 3D image represents the layer in the Z-stack. (E) Average biofilm thickness formed by NGJ1 strains, estimated from the CLSM images. Graphs show mean values ± standard error of three biological replicates. Asterisks * and ** indicate a significant difference at $P < 0.05$ and $P < 0.01$, respectively (estimated using one-way ANOVA). NS, nonsignificant.

strains in laboratory media, in the absence of fungal mycelia (Fig. 2C). Overall, our data suggest that NA promotes the mycophagous ability of *B. gladioli* strain NGJ1 on *R. solani*.

We observed that expression of the *nicT* gene (ACI79_RS00310), which encodes a major facilitator superfamily (MFS) transporter protein (potentially involved in NA uptake [16]), was upregulated in wild-type and Δ*nicC*/Δ*nicX* mutant bacterium during 48 h of interaction with *R solani* (Fig. S4). We anticipate that upregulation of NicT may facilitate the NGJ1 bacterium to take up NA during mycophagy.

**NA regulates motility and biofilm formation ability in *B. gladioli* strain NGJ1.** Considering that NA but not FA restored mycophagy in Δ*nicC* and Δ*nicX* mutant strains, we envisaged that NA may regulate mycophagy through modulation of bacterial physiological processes. Motility and biofilm formation ability play important roles during bacterial interaction with eukaryotic organisms (20). We observed that wild-type NGJ1 is proficient in swimming motility on laboratory plates (0.25% PDA), which gets enhanced in the presence of NA (Fig. 3A and B). The wild-type NGJ1 also produced biofilm on polystyrene plates and NA enhanced biofilm formation (Fig. 3C). Confocal laser scanning microscopic analysis further reflected that NA enhances biofilm formation on glass slides (Fig. 3D and E). Moreover, FA supplementation had no significant effect on the bacterial motility (Fig. 3A and B) and biofilm forming ability on polystyrene plates (Fig. 3C) as well as glass slides (Fig. 3D and E).

**Swimming motility as well as biofilm formation ability are compromised in ΔnicC and ΔnicX mutant strains.** We further tested the swimming motility and biofilm formation ability in ΔnicC and ΔnicX mutant strains. Compared to the wild-type NGJ1, ΔnicC and ΔnicX were less motile (Fig. 3A and B). Also, the mutants were compromised in biofilm formation on polystyrene plates (Fig. 3C) and glass slides (Fig. 3D and E). NA supplementation enhanced the motility (Fig. 3A and B) and biofilm formation ability in the mutant bacteria (Fig. 3C to E). On the other hand, supplementation of FA has no significant effect on the motility (Fig. 3A and B) and biofilm forming ability of ΔnicC/ΔnicX on polystyrene plates (Fig. 3C) as well as glass slides (Fig. 3D and E).

**nicR, encoding a MarR type of transcriptional repressor, is upregulated in ΔnicC and ΔnicX mutants of *B. gladioli* strain NGJ1.** It was intriguing why the mutation in NA catabolic genes leads to defects in bacterial motility and biofilm formation. Generally, NA catabolism is regulated by a MarR type of transcriptional repressor in bacteria which maintains the cellular NA pool (16). We observed that NGJ1 encodes NicR (ACI79_RS00315), a homolog of MarR and the expression of the gene was upregulated in ΔnicC and ΔnicX mutants during 48 h of confrontation with *R. solani*, compared to the wild-type NGJ1 (Fig. 4A). However, upon NA supplementation, *nicR* expression was comparable in ΔnicC, ΔnicX and wild-type NGJ1 during mycophagy (Fig. 4B). We anticipate that upon NA supplementation reduced expression of *nicR* restores mycophagous ability in the ΔnicC/ΔnicX mutant strains.

To investigate further, we created a ΔnicR mutant of NGJ1 and observed that it is compromised in mycophagous ability on *R. solani* (Fig. 4C, S5A). The supplementation of NA failed to enhance mycophagy in the ΔnicR (Fig. 4C, S5A), potentially due to the inability of the mutant bacterium to sense NA, as in case of ΔbpsR mutant of *B. bronchiseptica* (21). Notably, ΔnicR mutant bacterium was completely defective in swimming motility (Fig. 4D, S5 B); however, it had enhanced ability to form biofilm (Fig. 4E, S5 C and D), irrespective of the presence or absence of NA. Considering that ΔnicR bacterium was defective in motility, we analyzed the expression of *flhDC* genes that regulate flagellar biosynthesis (22, 23). The qRT-PCR analysis reflected that *flhC* (ACI79_RS06945) and *flhD* (ACI79_RS06940) genes were induced in wild-type as well as in ΔnicC but not in ΔnicR mutant strains during confrontation with *R. solani* (Fig. 4F). Moreover, expression of *flhC* and *flhD* genes were upregulated upon NA supplementation during confrontation of wild-type and ΔnicC strains with *R. solani* (Fig. 4G). On the other hand, the *flhDC* genes were not induced in ΔnicR during interaction with *R. solani*, even in the presence of NA (Fig. 4G). This suggested that ΔnicR mutant is compromised in flagellar biosynthesis. Transmission electron microscopic analysis reflected that wild-type and ΔnicC mutant bacteria possess a single polar flagellum while no flagella were seen in case of ΔnicR mutant during growth in laboratory media (Fig. 4H). Overall, our data suggest that impaired motility and excessive biofilm formation ability impart mycophagy defects in ΔnicR mutant.

## DISCUSSION

Mycophagy is an important trait that enables bacteria to survive under nutrient-limiting conditions by utilizing fungal biomass as a nutrient source. Previously we have demonstrated that a prophage tail-like protein (Bg_9562) is deployed by the *B. gladioli* strain NGJ1 to forage over fungi (3). In the present study, we demonstrate the importance of NA catabolism for mycophagous ability of *B. gladioli* strain NGJ1. Previous studies have shown that NA catabolism is important for bacterial virulence and growth under diverse environments (24, 25). Members of gammaproteobacteria (Pseudomonadales) and betaproteobacteria (Burkholderiales), including *Pseudomonas putida* and *Bordetella bronchiseptica* possess complete NA catabolic pathway (15, 16, 21). The operonic arrangement of NA catabolic (*nic*) genes differ between the *Pseudomonas* and *Bordetella* species. We observed that NA hydroxylase enzymes are tripartite (NicA, NicB1, NicB2) in *B. gladioli* (NGJ1 and BSR3) and *Bordetella* sp. (15). Similar to *P. putida*, *B. gladioli* strains harbor putative major facilitator superfamily transporter (NicT) at the *nic* locus, while *Bordetella* sp. harbors putative ATP binding cassette transporter, potentially for uptake of NA (15).

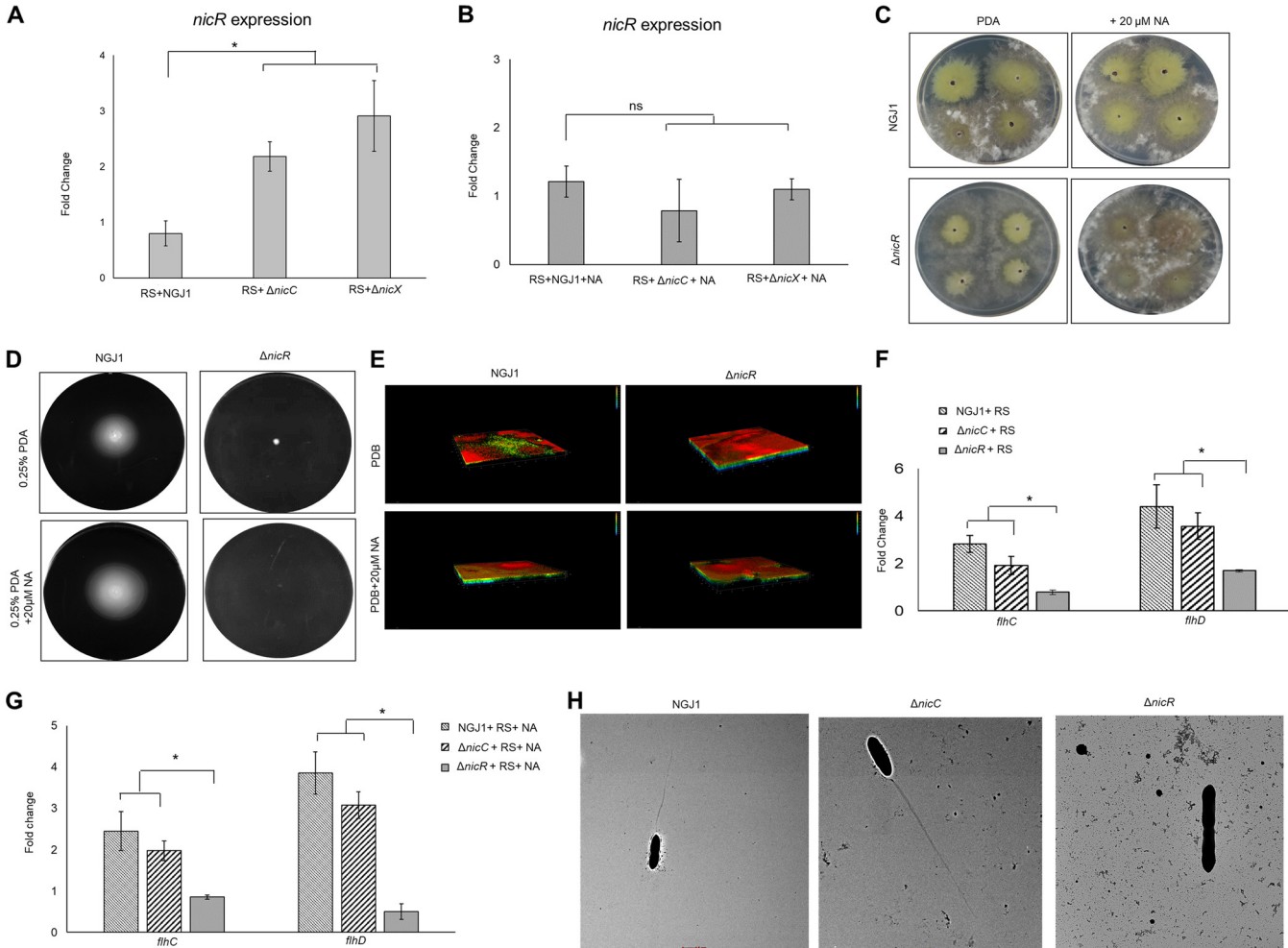

**FIG 4** *nicR*, a negative regulator of biofilm modulates bacterial mycophagy, swimming motility and biofilm formation in *B. gladioli* strain NGJ1. (A) qRT-PCR-based expression of *nicR* during 48 h of interaction of NGJ1 strains with *R. solani* (RS). (B) *nicR* expression upon NA supplementation during 48 h of mycophagous interaction. Representative images of (C) interaction of NGJ1 strains (10³ cells/mL density) with *R. solani* on PDA plates, at 7 DPI and (D) bacterial motility on semisolid swim (0.25% PDA) plates, after 16 h of growth. (E) Representative confocal laser-scanning microscopic images, showing biofilms formed by different NGJ1 strains on glass slides at the air-media interface upon staining with BacLight LIVE/DEAD stain, during growth at 72 h. Each 3D image represents the layer in the Z-stack. (F) qRT-PCR-based expression analysis of *flhC* and *flhD* genes during 48 h of mycophagous interaction. (G) The expression of *flhC* and *flhD* genes upon NA (20 μM) supplementation, during 48 h of mycophagy. (H) Representative images of transmission electron microscopy of flagella produced by NGJ1 strains. Scale: 1 μm. The experiments were independently repeated three times with a minimum of three technical replicates. The fold change was estimated by $2^{-\Delta Ct}$ method, using 16S rRNA of NGJ1 as the endogenous control. Graphs show mean values ± standard error of three biological replicates. Asterisk * indicates a significant difference at $P < 0.05$ (estimated using one-way ANOVA). NS, nonsignificant.

Our data reflected that NGJ1 is auxotrophic to NA, whereas supplementation with *R. solani* mycelial extract enables bacterial growth in minimal media without an additional carbon source. On the other hand, neither NA nor *R. solani* extract restored the growth of Δ*nicC*/Δ*nicX* mutants of NGJ1. This reinforced that NA catabolism is important for the growth of NGJ1 under carbon-deficient conditions. Notably, the Δ*nicC*/Δ*nicX* mutants were mycophagy defective and supplementation of NA promoted mycophagy. This reflects that NA is important for mycophagy and we anticipate that during confrontation NGJ1 may derive NA from *R. solani*. Considering that NA catabolism leads to the formation of fumaric acid (FA), a TCA cycle intermediate (16), we speculated that supplementation of FA may restore mycophagy in the Δ*nicC* and Δ*nicX* mutant strains. However, FA did not restore mycophagous ability in Δ*nicC*/Δ*nicX* mutants. This implies that during mycophagy NA might have additional regulatory functions, other than being a carbon source. NA catabolism is tightly regulated in bacteria by a MarR-type of transcriptional regulator i.e NicR in *P. putida* and BpsR (NicR homolog) in *B. bronchiseptica* (17, 21). Depending upon NA pool, the expression of

*nicC* and *nicR* genes is modulated (15). In wild-type *B. bronchiseptica* the *nicC* expression was enhanced in an NA concentration dependent manner, while the Δ*bpsR* mutant strain was unable to sense NA, exhibiting constant upregulation of *nicC* (15). We hypothesized that defects in *nicC*/*nicX* genes may alter the cellular NA pool in NGJ1. In this regard, we observed that compared to the wild-type NGJ1, *nicR* gene was upregulated in Δ*nicC* as well as Δ*nicX* mutants during confrontation with *R. solani*. On the other hand, supplementation of NA led to the basal level expression of *nicR* in Δ*nicC*/Δ*nicX* mutant strains. Considering the above, we anticipate that supplementation of NA restores NA pool, maintains basal level expression of *nicR* and therefore promotes mycophagy in Δ*nicC*/Δ*nicX* mutants.

It is known that bacterial motility and biofilm formation play an important role during host colonization. The fungal hypha serves as means for the long-distance movement of bacteria in rhizosphere (26). The soil-dwelling bacterium, *Bacillus subtilis* produces biofilms on the fungal mycelia of *Aspergillus niger* and *Agaricus bisporus* (27). We envisaged that mycophagous bacteria need to be efficient in establishing contact with fungi, colonizing and spreading over the mycelia. In this regard, we observed that wild-type NGJ1 is proficient, while Δ*nicC*/Δ*nicX* mutants are compromised in swimming motility as well as biofilm formation. The presence of NA but not FA enhanced the bacterial motility as well as biofilm forming ability and thereby promotes mycophagy in wild-type/Δ*nicC*/Δ*nicX* strains. It is noteworthy that Δ*nicR* was compromised in mycophagy and supplementation of NA did not restore mycophagous ability. We anticipate this is due to the inability of Δ*nicR* mutant of NGJ1 to sense NA, similar to the Δ*bspR* mutant of *Bordetella* (21). A previous study has shown that Δ*bpsR* mutant of *B. bronchiseptica* produces excessive biofilm (28). We observed that although the Δ*nicR* mutant of NGJ1 produced excessive biofilm but it was completely defective in swimming motility, irrespective of the presence or absence of NA in the media. Our data emphasize that the swimming motility in Δ*nicR* is compromised due to defects in flagellar biosynthesis. The transmission electron microscopic studies revealed that Δ*nicR* mutant was not able to produce flagella, while prominent flagellum was visible in wild-type NGJ1. The qRT-PCR analysis reflected that during mycophagy, the expression of *flhDC* genes that encode the master regulator of flagellar biosynthesis (23, 29, 30), was upregulated in wild-type NGJ1. On the other hand, the Δ*nicR* mutant was compromised in the expression of *flhDC* genes, even in the presence of NA. Bacterial flagella are important for adherence, maturation and dispersal of biofilm-forming cells to newer sites (31, 32). Generally, a transition from motility to biofilm is important for bacterial pathogenesis and an optimal biofilm consists of both motile and nonmotile cells (33, 34). Our data points out that due to the lack of flagella, Δ*nicR* mutant remains nonmotile and its ability to disperse from biofilm is compromised. Therefore, Δ*nicR* mutant exhibits reduced mycophagy compared to wild-type NGJ1.

Overall, our study suggests that NGJ1 requires NA for optimal growth. During mycophagous interaction, NGJ1 potentially senses fungal mycelia as a NA source and maintains the basal level expression of *nicR*. This in turn enables the bacterium to produce optimal biofilm, which is important for effective colonization on fungal mycelia. Further, swimming motility is an important trait that enables the NGJ1 bacteria to spread over fungi during mycophagy. We highlight that disruption of NA catabolic genes (*nicC* and *nicX*) alters the cellular NA pool in NGJ1, which in turn upregulates *nicR*, the negative regulator of biofilm formation. The upregulation of *nicR* impairs biofilm formation and bacterial motility, rendering mycophagy defects in the Δ*nicC*/Δ*nicX* mutant strains (Fig. 5).

## MATERIALS AND METHODS

**Growth conditions.** *B. gladioli* strains were grown on potato dextrose agar (PDA; HiMedia Laboratories, India) plates at 28°C. The *Tn* mutant strains of NGJ1 were grown on PDA supplemented with rifampicin and spectinomycin. The insertional mutants, Δ*nicC*, Δ*nicX* and Δ*nicR* were grown on PDA supplemented with rifampicin and kanamycin. *R. solani* AG1-IA strain BRS1 (35) was cultured on PDA plates at 28°C, and freshly collected sclerotia were used for mycophagy assays. *E. coli* strains were grown on LBA (Luria Bertani Agar media; HiMedia Laboratories, India) at 37°C. Whenever required, medium was

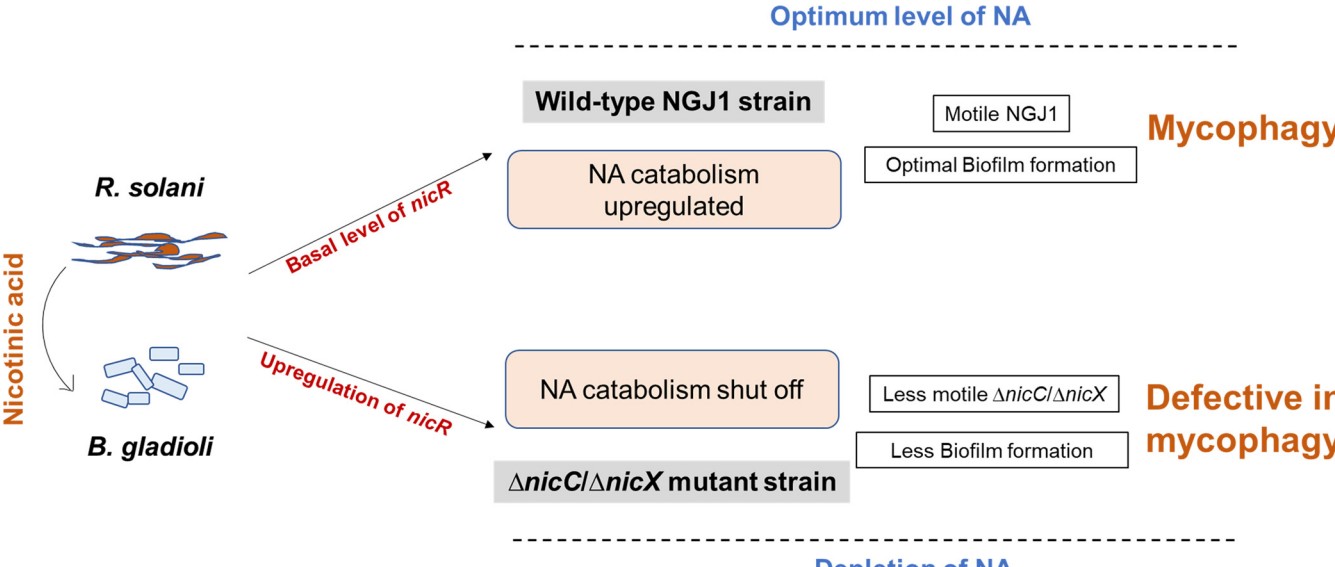

**FIG 5** NA catabolism modulates bacterial mycophagy in *B. gladioli* strain NGJ1. The wild-type NGJ1 bacterium senses the presence of NA from fungal (*R. solani*) mycelia/growth media and upregulates its NA catabolic pathway. This keeps the expression of *nicR* (a negative regulator of NA catabolism and biofilm formation) to the basal level. This enables the bacteria to form motile colonies and produce optimum biofilm which is important for mycophagy. On the other hand, disruption of *nicC/nicX* modulates NA pool, leading to upregulation of NicR which alters bacterial motility and ability to produce optimal biofilm, thereby negatively impacting mycophagy.

supplemented with appropriate concentration of antibiotics (kanamycin: 50 $\mu$g/mL; rifampicin:50 $\mu$g/mL; ampicillin: 50 $\mu$g/mL; and spectinomycin: 50 $\mu$g/mL). Various bacterial and fungal strains, as well as plasmids used in this study are enlisted in Table S2.

**Transposon mutagenesis in NGJ1.** The transposon, *Tn5gusA11* having a spectinomycin resistance marker was maintained in *E. coli* strain TP003 as a suicidal plasmid (36). Biparental mating between the donor *E. coli* TP003 and the recipient *B. gladioli* NGJ1 strain was performed, as described before (37). Briefly, a single colony of *E. coli* strain TP003 was mixed with the recipient *B. gladioli* NGJ1 cells on a piece of Hybond N Plus nylon membrane (Amersham Life Science, Buckinghamshire, UK). Upon incubation for 48 h at 28℃, the bacterial cells were scraped with a sterile toothpick, resuspended in 100 $\mu$L sterile water and plated on KBA (King's medium B Base; HiMedia, India) supplemented with rifampicin (50 $\mu$g/mL) and spectinomycin (50 $\mu$g/mL). The plates were incubated at 28℃ and positive conjugants were analyzed for *gus* ($\beta$-glucuronidase) activity, as described in (36). Briefly, bacteria were taken in a sterile microcentrifuge tube, mixed with 1 $\mu$L of X-gluc (5-bromo-4-chloro-3-indolyl-beta-d-glucuronic acid;100 mg/mL) and incubated at 37℃ for 30 min or until the appearance of blue color in the suspension.

**Screening of transposon-tagged mutants for mycophagy defects and identification of the transposon-tagged gene.** The transposon-tagged mutants were screened for defects in mycophagy against *R. solani* on PDA plates, as described before (3). The mutants compromised in mycophagy were selected. For identification of the disrupted gene, genomic DNA of the mutant bacterium was partially digested with SauIIIA (New England Biolabs, Inc. Ipswich, USA) and cloned into pBS-KS (+) vector, linearized with BamHI. The clone having the DNA fragment corresponding to spectinomycin marker of the transposon and the flanking sequence of NGJ1 (tagged-region) were selected on LBA plates containing ampicillin (50 $\mu$g/mL) and spectinomycin (50 $\mu$g/mL). The positive transformants were sequenced using M13 forward/reverse primers (Table S3). The sequences, thus obtained, were used for BLAST (Basic Local Alignment Search Tool) searches in the Burkholderia Genome Database (https://www.burkholderia.com) to identify the disrupted genes.

**Insertional mutagenesis through plasmid integration in the *nic* genes.** Partial fragments (250 to 300bp) of *nicC*, *nicX,* and *nicR* genes of NGJ1 were amplified from genomic DNA using gene specific primers (Table S3), cloned into pK18mob vector and electroporated into competent cells of wild-type NGJ1 using Gene pulsar XcellTm; Bio-Rad Laboratories, Inc., CA, USA, as described previously (4). The transformants were selected on rifampicin (50 $\mu$g/mL) and kanamycin (50 $\mu$g/mL) containing KBA plates. The mutants were confirmed by PCR using gene-specific flanking forward and M13 vector specific reverse primers.

**Bacterial-fungal confrontation assay.** The bacterial-fungal interaction was carried out as described previously (5, 38). Briefly, *R. solani* sclerotia were treated with 5 mL of $10^3$ cells/mL cell density of various NGJ1 strains for 4 h at 28℃. After washing, the bacterial-treated sclerotia were placed on PDA plates and the bacterial foraging over fungal mycelia was monitored at 3 and 7 DPI (days postinoculation). Wherever applicable, 20 $\mu$M nicotinic acid (NA; Sigma-Aldrich, Cat. no 72309) or fumaric acid (FA; Sigma-Aldrich, Cat. No. 47910) were supplemented onto the confrontation plates. The bacterial abundance on mycelial mass after 7 DPI was estimated by serial dilution plating and colony counting. Each experiment was repeated at least thrice with minimum 4 sclerotia per treatment being analyzed in each replicate.

**Fungal cell viability assay using MTT (3-[4,5-dimethylthiazol-2-yl]-2,5-diphenyltetrazolium bromide).** The cell viability assay was performed as described previously (5). Briefly, pregrown *R. solani* mycelial mass was treated with NGJ1 strains (1% inoculum of $10^9$ cells/mL) in liquid media (PDB) for 48 h at 28°C. *R. solani* mycelia without any bacterial treatment were used as a control. The bacterial cells loosely attached with fungal mycelia were washed (2 to 3 times) with PBS buffer (phosphate buffer saline; 10 mM, pH 7.4) and the fungal biomass was stained with 100 $\mu$L of MTT (HiMedia Laboratories, India; Catalog no. TC191) solution (5 mg/mL in PBS buffer). Upon incubation in the dark for 90 min, the appearance of a dark-colored formazon was observed. After washing the fungal mycelia with PBS buffer, bound formazon was extracted using absolute ethanol and quantified ($OD_{570}$) using a spectrophotometer (Bio-Rad Smart Spec-3000), as described previously (3).

**qRT-PCR based bacterial gene expression analysis during bacterial-fungal interaction.** The bacterial-treated fungal mycelia were filtered out from the liquid media using mira cloth and total RNA was isolated from the bacterial cells using RNeasy minikit (Qiagen), following manufacturer's instructions and as described before (5). One $\mu$g of total RNA was taken for cDNA synthesis, using Verso cDNA Synthesis kit (Thermo Fisher Scientific Inc.). The qRT-PCR was performed using gene specific primers (Table S3) and PowerUp SYBR green master mix (Applied Biosystems), on ABI 7900HT Real-time PCR (Applied Biosystems). The 16S rRNA of NGJ1 was used as the endogenous control and fold change in gene expression was estimated using $2^{-\Delta\Delta Ct}$ methods (39).

**Growth of NGJ1 strains in the presence of aromatic hydrocarbons (nicotinic acid, picolinic acid, isonicotinic acid) and the cell extract of *R. solani*.** For auxotrophy assay, serially diluted bacterial (NGJ1, $\Delta nicC$, $\Delta nicX$) cultures were grown in M9 minimal media ($FeSO_4$–$7H_2O$, $1.25 \times 10^{-4}$; $[NH_4]_2SO_4$, 0.5; $MgSO_4$–$7H_2O$, 0.05; $KH_2PO_4$, 3.4 g/L), with and without supplementation of 20 $\mu$M nicotinic acid/picolonic acid (Sigma-Aldrich, Cat. no P42800)/isonicotinic (Sigma-Aldrich, Cat. No. I17508)/cell free *R. solani* extract (RS) as sole carbon source, at 28°C. For the preparation of RS extract, 10 g (wet weight) of mycelial mass was thoroughly washed with water, ground into powder in a mortar and pestle using liquid nitrogen and dissolved in 10 mL sterile water. Upon filtration through a 0.22 $\mu$m filter, the solution was used as a cell extract.

**Bacterial swimming motility assay.** Semisolid plates were prepared with 0.25% agar in PDB and NA/FA (20 $\mu$M) were supplemented, wherever mentioned. Bacterial cultures ($OD_{600}$ = 0.5) were washed and resuspended in PBS. The bacterial cultures were spotted onto the center of the semisolid plates and incubated at 28°C for 16 h. The swim zone was monitored and the diameter (in cm) was recorded. The experiment was performed thrice independently with at least five technical replicates.

**Biofilm formation assay in 96 well-based polystyrene plates.** The biofilm formation assay in 96-well polystyrene plates was carried out as per previously described method (40) with slight modifications. Briefly, 100 $\mu$L bacterial cultures ($10^6$ cells/mL cell density) were grown in polystyrene 96-well plates with or without 20 $\mu$M NA/FA supplementation. After 72 h of static growth at 28°C, the medium was gently decanted and planktonic cells were washed off from the wells. The polystyrene surface attached bacterial cells forming biofilm were stained with 1% crystal violet for 10 min at room temperature. The bound crystal violet in the bacterial cells was solubilized using absolute methanol (100 $\mu$L) and quantified by measuring the absorbance at 540 nm. Each experiment was performed with five technical replicates and the average of three biological replicates was plotted as a graph.

**Visualization of biofilm under confocal laser scanning microscope.** The bacterial cultures ($10^6$ cells/mL cell density) were grown in PDB with or without supplementation of NA or FA (20 $\mu$M) in 50-mL falcon tubes containing a sterile glass slide half-dipped in the media and incubated for 72 h at 28°C, as described (40). Upon washing with sterile water, the bacterial biofilms formed at the air-media interface at the glass slide were stained using the BacLight LIVE/DEAD Bacterial Viability kit (L7012; Invitrogen, Eugene, OR, USA), as per the manufacturer's instructions. Slides were mounted with 25% glycerol and analyzed under 63× objective of confocal laser scanning microscope (Leica TCS-SP8). The 3D images were reconstructed using LAS AF Version: 2.6.0 build 7266 software. Biofilm thickness was quantified by measuring the average of horizontal (*xz*) optical sections (at 1 $\mu$m intervals) of the biofilm on the glass slides. Each experiment was performed with five technical replicates and the average of three biological replicates was plotted as a graph.

**Transmission electron microscopy for visualization of flagella.** Bacterial cultures were grown to log phase, cells were pelleted down and gently washed using 1 mM phosphate buffer (pH 7.2). The cells were fixed with 2.5% glutaraldehyde and 1.5% formaldehyde solution for overnight. Upon washing with phosphate buffer, cells were stained with phosphotungstic acid (PTA) and mounted on Carbon Film 200 Mesh Copper (Electron Microscopy Sciences). The grids were dried at room temperature and visualized under transmission electron microscope using SAIF-EM facility at All India Institute of Medical Sciences (AIIMS), New Delhi, India.

**Statistical analysis.** One-way analysis of variance was performed using Sigma Plot 11.0 software (SPSS, Chicago, IL, USA), using the Student–Newman–Keuls test considering $P \leq 0.001$, $P \leq 0.01$, and $P \leq 0.05$ as statistically significant. Where applicable, the significance is mentioned in the figure legend.

**Data availability.** All data related to the study are included in the article and supplemental material.

## SUPPLEMENTAL MATERIAL

Supplemental material is available online only.

**SUPPLEMENTAL FILE 1**, PDF file, 0.7 MB.

## ACKNOWLEDGMENTS

J.D. and S.K.Y. acknowledge SRF fellowship from Department of Biotechnology (DBT), Government of India. R.K. acknowledges the SRA fellowship from CSIR, Govt. of India. Subhash Chandra Yadav, Department of Anatomy is acknowledged for suggestions and facilitating microscopy study at Electron Microscope Facility (SAIF) AIIMS, India. NIPGR central instrumentation facilities for DNA sequencing, qRT-PCR, and confocal imaging are acknowledged. The authors are thankful to DBT-eLibrary Consortium (DelCON) for providing access to e-resources. The work has been supported by NIPGR core research grant. G.J. acknowledges Swarna Jayanti fellowship from SERB, Ministry of Science and Technology, Govt. of India and research fundings from department of biotechnology (DBT), Govt. of India. The funders had no role in study design, data collection, and analysis, decision to publish, or preparation of the manuscript.

G.J. planned and supervised the study. J.D. performed various microbiology and molecular biology experiments to establish the role of NA catabolism in mycophagy. R.K. created transposon mutants of NGJ1. S.K.Y. assisted in molecular cloning. G.J. and J.D. wrote the manuscript and all authors approved the manuscript.

The authors have no conflicts of interest to declare.

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

.ppat.1006019.

