## [Reviewer comments · Microbiology Spectrum]

Microbiology Spectrum

Nicotinic acid catabolism modulates bacterial mycophagy in *Burkholderia gladioli* strain NGJ1

Gopaljee Jha, Joyati Das, Rahul Kumar, and Sunil Yadav

Corresponding Author(s): Gopaljee Jha, National Institute of Plant Genome Research

Review Timeline:

Submission Date:	November 4, 2022
Editorial Decision:	January 5, 2023
Revision Received:	March 1, 2023
Accepted:	March 3, 2023

Editor: Cezar Khursigara

Reviewer(s): Disclosure of reviewer identity is with reference to reviewer comments included in decision letter(s). The following individuals involved in review of your submission have agreed to reveal their identity: Yuji Morita (Reviewer #1)

Transaction Report:

DOI: <https://doi.org/10.1128/spectrum.04457-22>

January 5, 2023

Dr. Gopaljee Jha
National Institute of Plant Genome Research
Plant microbe interactions lab
Aruna Asaf Ali Marg
New Delhi, Delhi 110067
India

Re: Spectrum04457-22 (Nicotinic acid catabolism modulates bacterial mycophagy in *Burkholderia gladioli* strain NGJ1)

Dear Dr. Gopaljee Jha:

Link Not Available

Sincerely,

Cezar Khursigara

Journals Department
Reviewer comments:

Reviewer #1 (Comments for the Author):

It is well documented. I have only minor comments.

1) Is it necessary that nucleotide sequences of the *nic* genes in *B. gladioli* NGJ1 goes to GenBank/DDBJ/EMBL?

2)Line 174: Is a preposition necessary "between" mutant and "*B. gladioli*"?

Reviewer #2 (Comments for the Author):

The manuscript by Das and co-workers reports a detailed study that 'Nicotinic acid catabolism modulates bacterial mycophagy in Burkholderia gladioli strain NGJ1' involves the characterization of nicotinic acid catabolic pathway genes in the mycophagous bacteria, Burkholderia gladioli NGJ1. Here, using mutational studies the authors have demonstrated that NA catabolic pathway genes, nicC and nicX are responsible for the fungal eating property of the bacterium. They have shown that defects in NA catabolism impact the nicR transcriptional regulator which in turn modulates bacterial ability to form biofilm and motility. The manuscript is of broader interest and highlights new insights into the biology of mycophagy. It connects mycophagy with the bacterial ability to form biofilm and motility and emphasizes the involvement of NA in these processes. I have pointed out some of the comments and suggestions towards strengthening of the manuscript.

Major Comments:

The authors propose that NGJ1 utilizes NA during mycophagy. However, it does not discuss how the bacteria uptakes NA from the extracellular milieu. Various possibilities of NA uptake should be discussed. Moreover, expression analysis of genes associated with NA uptake during mycophagy should be helpful in this regard.

1. In Fig 1 D, the growth difference between NGJ1 and mutant strains on NA and R. solani extract is not clear. The experiment should be repeated and a better-quality image should be provided.

2. Why NA supplementation restores mycophagy in Δ nicC and Δ nicX is not clear.

3. The effect of fumaric acid, the end product of NA catabolism on biofilm formation/motility of wild-type and mutant strains should be tested to clarify whether NA has a regulatory role or whether the bacterium utilizes as a carbon source to modulate these processes.

4. The authors show that the biofilm formation ability and motility in the NGJ1 bacterium are enhanced by NA, whereas in the Δ nicR mutant these phenomena are not modulated by NA. The gene expression analysis of the motility and biofilm-associated genes in the presence and absence of NA should be helpful to strengthen the data.

Other Comments:

Line 182- Figure 4 A the expression analysis of nicR in Δ nicX mutant is missing

Line 324- Mention locus ID of nicC, nicX and nicR genes

Line 347- Source of MTT used?

Line 379- It is not clear R. solani extract was prepared, this needs to be clearly described in the method section and the source of NA and other compounds used, should be provided.

Line 391- remove, 'approximately'

Staff Comments:

Preparing Revision Guidelines

Please return the manuscript within 60 days; if you cannot complete the modification within this time period, please contact me. If you do not wish to modify the manuscript and prefer to submit it to another journal, please notify me of your decision immediately so that the manuscript may be formally withdrawn from consideration by Microbiology Spectrum.

Reviewer #1 (Comments for the Author):

It is well documented. I have only minor comments.

Response: Thank you very much for appreciating our work. We have now incorporated the suggested changes.

Query: 1) Is it necessary that nucleotide sequences of the *nic* genes in *B. gladioli* NGJ1 goes to GenBank/DDBJ/EMBL?

Response: Kindly note, the nucleotide sequences of the *nic* genes in *B. gladioli* strain NGJ1 are publicly accessible from the Burkholderia genome database. We have included the locus ID and protein IDs in Supplementary Table S1 and have also mentioned them in the manuscript.

Query: 2) Line 174: Is a preposition necessary "between" mutant and "*B. gladioli*"?

Response: We have incorporated the suggested change.

Reviewer #2 (Comments for the Author):

The manuscript by Das and co-workers reports a detailed study that 'Nicotinic acid catabolism modulates bacterial mycophagy in Burkholderia gladioli strain NGJ1' involves the characterization of nicotinic acid catabolic pathway genes in the mycophagous bacteria, Burkholderia gladioli NGJ1. Here, using mutational studies the authors have demonstrated that NA catabolic pathway genes, *nicC* and *nicX* are responsible for the fungal eating property of the bacterium. They have shown that defects in NA catabolism impact the *nicR* transcriptional regulator which in turn modulates bacterial ability to form biofilm and motility. The manuscript is of broader interest and highlights new insights into the biology of mycophagy. It connects mycophagy with the bacterial ability to form biofilm and motility and emphasizes the involvement of NA in these processes. I have pointed out some of the comments and suggestions towards strengthening of the manuscript.

Response: Thank you very much for critically reading our manuscript and providing valuable suggestions to improve it further. We have revised the manuscript as per your suggestions.

Major Comments:

Query: The authors propose that NGJ1 utilizes NA during mycophagy. However, it does not discuss how the bacteria uptakes NA from the extracellular milieu. Various possibilities of NA uptake should be discussed. Moreover, expression analysis of genes associated with NA uptake during mycophagy should be helpful in this regard.

Response: Thank you very much for pointing this out. In the previous version of our manuscript, we had briefly mentioned that the *nic* locus encodes a putative NA transporter i.e NicT, a MFS transporter. We have now included qRT-PCR data to show that *nicT* is upregulated during mycophagous interaction of NGJ1 with *R. solani* (Figure S4). We hypothesize that NicT may facilitate NGJ1 to uptake NA.

Query: 1. In Fig 1 D, the growth difference between NGJ1 and mutant strains on NA and *R. solani* extract is not clear. The experiment should be repeated and a better-quality image should be provided.

Response: We have repeated the experiment and have provided a better quality image (Figure 1D). Kindly note, in the given figure panel, the growth of NGJ1 is reflected on M9 minimal media that lack a carbon source. Due to this, the growth is limited and upon the addition of *R. solani* mycelial extract, the growth is promoted.

Query: 2. Why NA supplementation restores mycophagy in $\Delta nicC$ and $\Delta nicX$ is not clear.

Response: Thank you for pointing this out. Kindly note, our data suggest that NA has regulatory functions viz. modulation of bacterial motility and biofilm formation ability and thereby regulating mycophagy. We emphasize that mycophagy defects in $\Delta nicC/\Delta nicX$ mutants are due to upregulated expression of *nicR*, the negative regulator of bacterial biofilm. Upon NA supplementation, the *nicR* expression is reduced to the basal level, comparable to that of wild-type NGJ1, and hence the motility, biofilm formation and mycophagy are restored in the $\Delta nicC/\Delta nicX$ mutants.

We have now included qRT-PCR data to show that the expression of *nicT* gene, encoding a potential NA transporter, gets upregulated in both wild-type as well as the $\Delta nicC/\Delta nicX$ mutant strains during interaction with *R. solani*. This suggests that although the $\Delta nicC/\Delta nicX$ mutant bacteria are defective in NA catabolism but they are proficient in NA uptake. Hence upon NA supplementation, the *nicR* expression is reduced in the $\Delta nicC/\Delta nicX$ and hence they are able to balance motility and form an optimum biofilm, therefore mycophagy is restored.

Query: 3. The effect of fumaric acid, the end product of NA catabolism on biofilm formation/motility of wild-type and mutant strains should be tested to clarify whether NA has a regulatory role or whether the bacterium utilizes as a carbon source to modulate these processes.

Response: As suggested, we have included data on fumaric acid (FA) supplementation on bacterial motility and biofilm formation. The data has been included in Figure 3 and it emphasizes that supplementation of FA has no significant role in the biofilm-forming ability and motility of the NGJ1 strains (wild-type and $\Delta nicC/\Delta nicX$ mutant). On the other hand, supplementation of NA promoted motility and biofilm formation in NGJ1 strains.

Query: 4. The authors show that the biofilm formation ability and motility in the NGJ1 bacterium are enhanced by NA, whereas in the Δ nicR mutant these phenomena are not modulated by NA. The gene expression analysis of the motility and biofilm-associated genes in the presence and absence of NA should be helpful to strengthen the data.

Response: Thank you for the kind suggestion. We have now included the expression of the *flhDC* genes of NGJ1, during mycophagy, without supplementation of NA (Figure 4 F) and upon NA supplementation (Figure 4G). The data reflected that *flhDC* genes are upregulated in wild-type bacteria in the presence of NA, whereas the Δ nicR mutant is insensitive to NA, as NA fails to induce the expression of *flhDC* genes in Δ nicR. This supports our finding that NA modulates bacterial motility and biofilm formation ability in wild-type NGJ1 and Δ nicR is insensitive to NA.

Other Comments:

Query: Line 182- Figure 4 A the expression analysis of nicR in Δ nicX mutant is missing

Response: We have now included the expression analysis of *nicR* in the Δ nicX mutant.

Query: Line 324- Mention locus ID of nicC, nicX and nicR genes

Response: Thank you for pointing this out. We have now mentioned the locus ID of the nic genes as per Burkholderia Genome Database in the supplementary table (Table S1) and main text.

Query: Line 347- Source of MTT used?

Response: Thank you for pointing this out. We have now mentioned the source of MTT used in the experiment.

Query: Line 379- It is not clear *R. solani* extract was prepared, this needs to be clearly described in the method section and the source of NA and other compounds used, should be provided.

Response: Thank you for pointing this out. We have elaborated on the methodology used for the extraction of *R. solani* mycelial extract, in the method section.

Query: Line 391- remove, 'approximately'

Response: As suggested, we have now removed the word.

March 3, 2023

Dr. Gopaljee Jha
National Institute of Plant Genome Research
Plant microbe interactions lab
Aruna Asaf Ali Marg
New Delhi, Delhi 110067
India

Re: Spectrum04457-22R1 (Nicotinic acid catabolism modulates bacterial mycophagy in Burkholderia gladioli strain NGJ1)

Dear Dr. Gopaljee Jha:

Your manuscript has been accepted, and I am forwarding it to the ASM Journals Department for publication. You will be notified when your proofs are ready to be viewed.

Sincerely,

Cezar Khursigara
Editor, Microbiology Spectrum
